# The effect of preoperative behaviour change interventions on pre- and post-surgery health behaviours, health outcomes, and health inequalities in adults: A systematic review and meta-analyses

**Mackenzie Fong**[1,2]*, **Eileen Kaner**[1,2], **Maisie Rowland**[2], **Henrietta E. Graham**[3], **Louise McEvoy**[2], **Kate Hallsworth**[4], **Gabriel Cucato**[5], **Carla Gibney**[2], **Martina Nedkova**[2], **James Prentis**[6], **Claire D. Madigan**[3]

1 NIHR Applied Research Collaboration, North East and North Cumbria, United Kingdom, 2 Population Health Sciences Institute, Newcastle University, Newcastle Upon Tyne, United Kingdom, 3 Centre for Lifestyle Medicine and Behaviour (CLiMB), The School of Sport, Exercise and Health Sciences, Loughborough University, Loughborough, United Kingdom, 4 NIHR Newcastle BRC, Newcastle upon Hospitals NHS Foundation Trust, Newcastle Upon Tyne, United Kingdom, 5 Faculty of Health and Life Sciences, Northumbria University, Newcastle Upon Tyne, United Kingdom, 6 Department of Perioperative and Critical Care Medicine, Freeman Hospital, Newcastle Upon Hospitals NHS Foundation Trust, Newcastle Upon Tyne, United Kingdom

* mackenzie.fong@newcastle.ac.uk

## Abstract

### Background

Prehabilitation interventions are being delivered across surgical specialities to improve health risk behaviours leading to better surgical outcomes and potentially reduce length of hospital stay. Most previous research has focused on specific surgery specialities and has not considered the impact of interventions on health inequalities, nor whether prehabilitation improves health behaviour risk profiles beyond surgery. The aim of this review was to examine behavioural Prehabilitation interventions across surgeries to inform policy makers and commissioners of the best available evidence.

### Methods and findings

A systematic review and meta-analysis of randomised controlled trials (RCTs) was conducted to determine the effect of behavioural prehabilitation interventions targeting at least one of: smoking behaviour, alcohol use, physical activity, dietary intake (including weight loss interventions) on pre- and post-surgery health behaviours, health outcomes, and health inequalities. The comparator was usual care or no treatment. MEDLINE, PubMed, PsychINFO, CINAHL, Web of Science, Google Scholar, Clinical trials and Embase databases were searched from inception to May 2021, and the MEDLINE search was updated twice, most recently in March 2023. Two reviewers independently identified eligible studies, extracted data, and assessed risk of bias using the Cochrane risk of bias tool. Outcomes

**Data Availability Statement:** All relevant data are within the manuscript and its Supporting Information files.

**Funding:** This study is funded by the National Institute for Health and Care Research (NIHR) Applied Research Collaboration (ARC) North East and North Cumbria (NIHR200173). The views expressed are those of the author(s) and not necessarily those of the NIHR or the Department of Health and Social Care." The funders had no role in study design, data collection and analysis, decision to publish, or preparation of the manuscript.

**Competing interests:** No authors have competing interest.

were length of stay, six-minute walk test, behaviours (smoking, diet, physical activity, weight change, and alcohol), and quality of life.

Sixty-seven trials were included; 49 interventions targeted a single behaviour and 18 targeted multiple behaviours. No trials examined effects by equality measures. Length of stay in the intervention group was 1.5 days shorter than the comparator (n = 9 trials, 95% CI -2.6 to -0.4, p = 0.01, $I^2$ 83%), although in sensitivity analysis prehabilitation had the most impact in lung cancer patients (-3.5 days). Pre-surgery, there was a mean difference of 31.8 m in the six-minute walk test favouring the prehabilitation group (n = 19 trials, 95% CI 21.2 to 42.4m, $I^2$ 55%, P <0.001) and this was sustained to 4-weeks post-surgery (n = 9 trials, mean difference = 34.4m (95%CI 12.8 to 56.0, $I^2$ 72%, P = 0.002)). Smoking cessation was greater in the prehabilitation group before surgery (RR 2.9, 95% CI 1.7 to 4.8, $I^2$ 84%), and this was sustained at 12 months post-surgery (RR 1.74 (95% CI 1.20 to 2.55, $I^2$ 43%, $Tau^2$ 0.09, p = 0.004)There was no difference in pre-surgery quality of life (n = 12 trials) or BMI (n = 4 trials).

## Conclusions

Behavioural prehabilitation interventions reduced length of stay by 1.5 days, although in sensitivity analysis the difference was only found for Prehabilitation interventions for lung cancer. Prehabilitation can improve functional capacity and smoking outcomes just before surgery. That improvements in smoking outcomes were sustained at 12-months post-surgery suggests that the surgical encounter holds promise as a teachable moment for longer-term behavioural change. Given the paucity of data on the effects on other behavioural risk factors, more research grounded in behavioural science and with longer-term follow-up is needed to further investigate this potential.

## Introduction

Each year approximately 310 million major operations are performed worldwide [1]. Major surgery imposes significant metabolic stress on patients [2], and complication rates following major surgery remain around 20% [3]. A recent paradigm shift has steered focus towards a proactive model of optimising patients' health and function in the weeks to months leading up to surgery (i.e., the pre-operative period) to improve resilience to surgical stressors and facilitate recovery—a practice that has come to be known as 'prehabilitation'. As well as better perioperative outcomes for individual patients, prehabilitation may also improve surgery throughput and resource efficiency by promoting earlier discharge from hospital. This is a priority for governments internationally as they continue to deal with surgical waiting lists that have been exacerbated by the Covid-19 pandemic [4,5].

While prehabilitation can involve medical optimisation (e.g., correction of anaemia, medication adjustment [6]) it may also involve behaviour change, leveraging its capacity as a 'teachable moment' where patients may be more motivated to adopt risk-reducing health behaviours [7]. Most behavioural programmes under study in previous literature invariably include an exercise component to improve cardiorespiratory fitness and muscular conditioning. There is also evidence that other behavioural health risk factors including poor diet [8,9] (and resulting excess weight [10]), smoking [11–13], and alcohol use [14–16] may heighten the risk of poorer perioperative outcomes, although prehabilitation programmes addressing these behaviours are relatively understudied.

Behavioural prehabilitation offers the possibility of sustained health behaviour change and, subsequently, public health gain [17] given 1) the large number of operations performed, 2) the high prevalence of behavioural health risk factors in surgical populations [18], and 3) that smoking, risky alcohol use, inadequate physical activity, poor diet (and resulting excess weight) are leading causes of preventable ill-health globally [19,20], including conditions that are commonly managed with surgery, e.g. cancers [21]. Longer elective surgery waiting lists in the wake of the COVID-19 pandemic [22] have prompted the Royal Colleges of Anaesthetists and Surgeons and the Centre for Perioperative Care to call for surgery waiting lists to be turned into 'preparation lists' [23], with behavioural modification being recognised as an important component [23]. Behavioural modification interventions in surgical populations also have the potential to reduce health inequalities because people from lower socioeconomic backgrounds are disproportionately represented in hospital and surgery populations, and tend to have more prevalent and multiple health risk behaviours [24,25] and related non-communicable disease [24,25]. Therefore, providing behavioural prehabilitation to patients on waiting lists may be more effective at reaching those from lower socioeconomic groups compared to more universal interventions.

Reviews (and a recent umbrella review [26]) of prehabilitation interventions addressing health risk behaviours have been conducted. However, these have focused on just one or two health behaviours [27–29], have been restricted to digital interventions [30] or have focused on special clinical groups (i.e., bariatric surgery [30], patients with alcohol dependency [26]). Further, no previous reviews have sought to examine the impact of prehabilitation interventions by health inequalities. The primary aim of this review was to examine the effect of behavioural Prehabilitation interventions that target physical activity, diet (including weight loss), alcohol use and smoking on pre- and post- surgery health outcomes and health behaviours across surgery specialities. The secondary aim was to examine these outcomes across the socioeconomic spectrum.

## Methods

The systematic review was registered on Prospero (CRD42021249265). This review is reported according to the Preferred Reporting Items for Systematic Reviews and Meta-Analyses (PRISMA) (see S1 File) [31,32]. Protocol amendments and rationale are presented in S1 Table in S1 File.

### Study inclusion criteria

RCTs with adult participants (≥18 years) that evaluated a prehabilitation intervention initiated before surgery that targeted dietary intake, weight loss, physical activity, alcohol use and/or smoking behaviours were eligible for inclusion. We included physical activity interventions that targeted any subtype of physical activity e.g. (both supervised and non-supervised) exercise [33] as in theory, all subtypes have the same aim of increasing cardio-respiratory fitness to improve the resilience for surgery, and can also promote regular, physical activity over the longer-term. There were no limitations on setting (e.g., hospital-based, home-based) or mode of delivery (e.g., digital or in-person). Trials were included if the comparator group received usual care or an intervention that did not focus on behaviour change. Table 1 details the full inclusion and exclusion criteria.

### Searches

We conducted a search of the following databases from inception to May 2021: MEDLINE, PubMed, PsychINFO, CINAHL, Web of Science, Google Scholar, Clinical Trials and Embase.

**Table 1. Review inclusion and exclusion criteria.**

| | Inclusion | Exclusion |
|---|---|---|
| Population | Adults (aged 18 years or over) who have been identified as requiring surgery/surgical procedure and referred to hospital care; those experiencing pre-assessment and awaiting treatment | Alcohol dependent patients; patients awaiting bariatric (weight loss) surgery; patients with major psychiatric conditions/lacking capacity |
| Intervention | Interventions initiated before surgery and delivered in any setting e.g., primary care, community, hospital; interventions where aim is to improve one or more health behaviour/s i.e., dietary intake (including weight loss), alcohol/tobacco use, physical activity/sedentary behaviour, regardless if explicitly acknowledged as an intervention target; any mode of delivery e.g., digital or in-person. | Exercise interventions that aim to strengthen a specific set of muscles rather than promote whole-body/general resistance training e.g., inspiratory muscle training; interventions where the diet component only involves dietary supplementation e.g., protein drinks |
| Comparator | Usual care, no intervention, or an intervention that does not aim to modify health behaviours (diet, physical activity, tobacco, or alcohol). | No comparator group |
| Outcome | Health behaviours i.e., one or more of dietary intake, physical activity/ sedentary behaviour, smoking/tobacco use, alcohol use or anthropometric outcomes e.g., weight; functional capacity (i.e., 6-minute walk test); health care service use (length of stay); quality of life | Anthropometric or dietary outcomes where the intervention aimed to promote weight maintenance/gain (not weight loss) |
| Study design | Individually or cluster randomised controlled trials. | Articles not published in English |

The main search terms were: health behaviours AND trial AND surgery. S1 File details the search for MEDLINE. Reference lists of included trials were hand searched to check for any additional studies not identified by the main searches. An updated search was conducted in Medline from 2021 to August 2022 and again in March 2023. We only updated Medline due to the likelihood that most abstracts would be found there.

### Data extraction

Results were uploaded to Rayyan [34], a software platform used for screening, and duplicates were removed. Two independent reviewers screened study titles, abstracts and full texts (from among MF, MR, LE, KH, GC, MN, CG). If there were disagreements these were resolved by consensus or by a third reviewer (from among EK and JP). Full texts were uploaded to Covidence systematic review software [35]. All decisions of inclusion or exclusion were automatically recorded in Covidence, and reviewers were blinded to each other's decisions. Data about study characteristics were extracted from among five authors (from among MF, MR, LM, CG, MN) (see S2 Table in S1 File) and three authors independently extracted outcome data. We contacted four authors [36–39] of included trials for further information about outcome data. For one trial the data were not accessible [39] and for one trial [37] the authors provided mean changes and standard deviations. The author of one study [40] was contacted to clarify intervention components. As no response was received this study was not included in this review.

### Outcomes, summary measures and synthesis of results

The aim of this review was to provide an overview of the effect of Prehabilitation interventions initiated before surgery that address health risk behaviours and was exploratory in nature. Therefore, there were several outcomes of interest all of which held equal value: healthcare usage (length of stay (LOS)); behavioural outcomes (diet, anthropometry, physical activity, smoking, alcohol use), functional outcomes (functional capacity i.e., 6 Minute Walk Test (6MWT)), and quality of life (QoL). In the narrative synthesis, only findings of definitive trials (i.e., not pilot/feasibility studies) were presented; these studies are more likely to be powered to detect between-group differences. Outcomes assessed after the prehabilitation intervention and just prior to surgery (herein referred to as the 'pre-surgery' timepoint) were extracted

where reported. Post-surgery outcomes were only extracted if both groups received the same treatment after surgery i.e., both the intervention and control group did or did not receive rehabilitation after surgery. To examine the impact of interventions on health inequalities, we extracted data about income, ethnicity, employment, education, deprivation, and sex.

### Meta-analyses

Many studies reported LOS, 6MWT, QoL, BMI, and smoking behaviour and were synthesised in a meta-analysis. We took a pragmatic approach and identified the most common measurement for each outcome with a minimum of three studies. Post-surgery outcomes were assessed at multiple timepoints. As they are likely to change over time, we analysed outcomes at discrete timepoints (e.g., 4 weeks, 1-year) rather than analysing aggregated outcomes. All analyses were conducted using Review Manager 5.4 [41]. Random effects models were used as the diversity of intervention components and comparator conditions meant that treatment effects were expected to differ. Many studies did not report mean changes and, therefore, we checked there were no data to suggest that baseline measures differed between groups, and then entered follow-up data.

Trials that reported LOS as a mean (SD) were included in the meta-analysis and a pooled mean difference was calculated. Mean difference was calculated for 6MWT outcomes at pre-surgery and post-surgery. $I^2$ were reported to quantify heterogeneity and $Tau^2$ to report between study variances. We generated funnel plots to evaluate small study effects (an indication of publication bias). We conducted the same analysis of QoL and BMI change at pre-surgery only, as there were not enough studies post-surgery. A pooled risk ratio was calculated for smoking cessation pre-operatively and at 12 months using random effects models. Three trials [42–44] measured smoking abstinence across pre-surgery and three weeks post-surgery and were included in the analysis as above.

**Risk of bias.** Two authors (CM and HG) independently assessed the risk of bias for each included study, using the Cochrane Risk of Bias Tool v2 (ROB2) [45]. For incomplete outcome data, a high risk of bias was defined as ≥20% attrition. We resolved disagreements by discussion or consulting a third review author (EK).

## Results

There were 67 unique trials that met our eligibility criteria. Some trials had published more than one paper and, therefore, the total number of papers that were eligible for inclusion was 74. **Fig 1** shows the PRISMA flow diagram.

### Study characteristics

Most of the trials (S2 Table in S1 File) were conducted in Canada (n = 15) [27,46–63], UK (n = 11) [64–74] and USA (n = 9) [39,75–83]. The number of participants included ranged from 3 to 761 (median = 88). Two trials included only women [44,82], two included only men [60,61,78] and two did not record sex [83,84]. On average the percentage of women was 42.3%, and the average age (of those that reported mean age) was 62.8 years. Only 13 studies reported a measure of socioeconomic status [37,43,46,59–61,70,78,79,85–89] and these included education, income and index of multiple deprivation (a UK measure). Only seven trials reported ethnicity [60,61,70,72,78,80,82,88]. The trial designs included RCTs (n = 49; 73.1%) [36–39,42–44,46–49,52,54–57,59,63–67,69,75–78,81,82,84–107], feasibility/pilot RCTs (n = 18; 26.9.%) [50,51,53,58,60–62,68,70–74,80,83,108–111], including one pilot/feasibility cluster RCT [79].

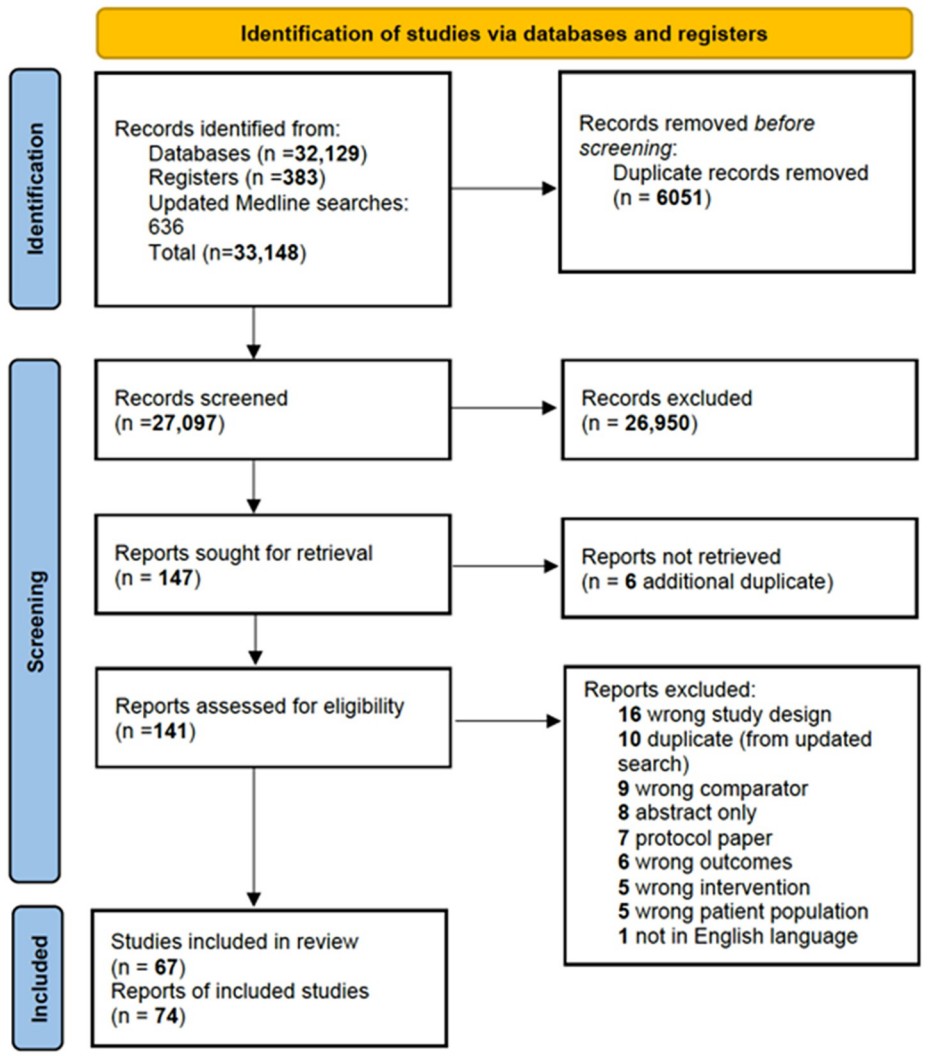

**Fig 1. PRISMA flow diagram.**

## Intervention characteristics

There were 18 (26.9%) interventions that targeted multiple behaviours [36,37,47–52,57,58,62, 67,69,74,76,78,81,87,94,111] and 49 (73.1%) that focused on a single behaviour [38,39,42–44, 46,53–56,59–61,64–66,68,70–73,75,77,79,80,82–86,88,90–93,95–110] amongst the behaviours of interest. Most interventions targeted physical activity (n = 34; 50.7%) [38,39,46,53,56,60,61, 64–66,70,71,73,75,77,80,82–84,86,88,90–93,95–98,100–102,105,108–110], physical activity and dietary intake (n = 11; 16.4%) [37,47,52,57,58,62,67,74,76,78,81,87,94] and smoking behaviour (n = 13; 19.4%) [42–44,54,55,59,63,68,79,99,103,104,106,107]. One intervention focused solely on alcohol use [72] and one on dietary intake [85]. Four interventions focused on changing all four behaviours (physical activity, dietary intake, smoking and alcohol use) [36,48,69,111]. The duration of intervention ranged from one session to nine months, with a median duration of four weeks, although duration was not reported in 13 studies. In many cases, it was difficult to ascertain up to when the intervention was delivered in relation to surgery e.g., the intervention was delivered up to the day of surgery, or the intervention was completed a week prior to

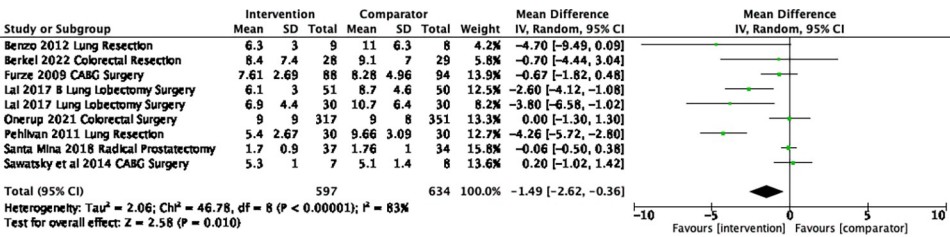

**Fig 2. Mean difference in the length of stay (days).**

surgery. As a proxy we used the time the outcome data was collected pre-surgery. Intervention characteristics are presented in S3 Table in S1 File.

## Outcomes

There were 18 trials that did not specify when the pre-surgery outcome was measured other than stating 'pre-surgery'. The other trials measured pre-surgery outcomes between one week before surgery, up to the day of admission for surgery. S1 and S2 Figs in S1 File comprehensively depict the behaviours targeted, the outcomes assessed and at which time point for all studies.

**Length of stay.** There were 44 trials [36–39,42,44,46,48,49,51,54,56–62,64–67,70–73,75,77,81,84,88,91,92,94–97,99–102,104,108–110,112]) that reported LOS. Of the 32 trials that assessed between-group differences, seven trials (21.9%) [46,64,84,91,95,96,100] found that the Prehabilitation group had a shorter LOS than the comparator group. Only nine trials reported length of hospital stay as means (SD) and were included in a meta-analysis [60,62,66,88,92,95,96]. There was a mean difference of -1.5 days (95%CI –2.6 to -0.4, $I^2$ 83%, Tau$^2$ 2.06, p = 0.01) in favour of the intervention group (**Fig 2**). In a post-hoc sensitivity analysis focusing on the type of surgery, only lung cancer surgery (n = 4) was associated with a significant difference of -3.6 days (95% CI -4.5 to -2.6, p<0.001, $I^2$ 0%). As there were less than two of each other surgical specialty we combined them and there was no difference in LOS (-0.1 days, 95% CI -0.5 to 0.3, p = 0.59, $I^2$ 0%).

**Functional capacity.** Thirty-one trials [36–38,47–49,52,53,56–58,60,62,70,71,74,80,83,85,86,91,93–97,102,105,108–110,112] assessed functional capacity, predominately using the 6MWT. There were 19 trials [37,47–50,57,58,60,62,70,71,74,85,91,94–96,105,108] (n = 1285) included in the meta-analyses of 6MWT pre surgery as they reported data that were able to be synthesized (**Fig 3**). A significant mean difference of 31.8m (95% CI

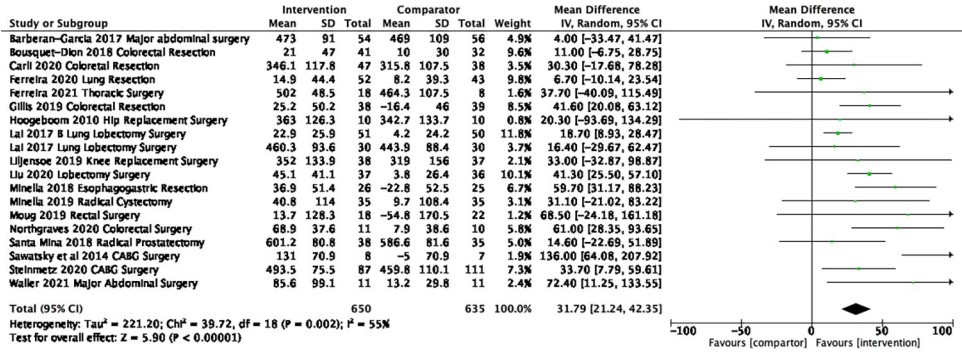

**Fig 3. Mean difference in 6MWT from baseline to pre-surgery.**

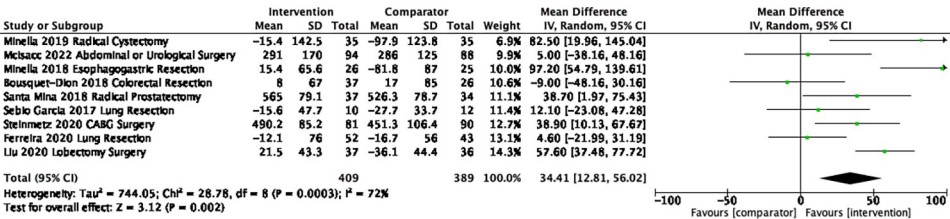

| Study or Subgroup | Intervention Mean | SD | Total | Comparator Mean | SD | Total | Weight | Mean Difference IV, Random, 95% CI |
|---|---|---|---|---|---|---|---|---|
| Minella 2019 Radical Cystectomy | −15.4 | 142.5 | 35 | −97.9 | 123.8 | 35 | 6.9% | 82.50 [19.96, 145.04] |
| McIsacc 2022 Abdominal or Urological Surgery | 291 | 170 | 94 | 286 | 125 | 88 | 9.9% | 5.00 [−38.16, 48.16] |
| Minella 2018 Esophagogastric Resection | 15.4 | 65.6 | 26 | −81.8 | 87 | 25 | 10.0% | 97.20 [54.79, 139.61] |
| Bousquet–Dion 2018 Colorectal Resection | 8 | 67 | 37 | 17 | 85 | 26 | 10.6% | −9.00 [−48.16, 30.16] |
| Santa Mina 2018 Radical Prostatectomy | 565 | 79.1 | 37 | 526.3 | 78.7 | 34 | 11.1% | 38.70 [1.97, 75.43] |
| Sebio Garcia 2017 Lung Resection | −15.6 | 47.7 | 10 | −27.7 | 33.7 | 12 | 11.4% | 12.10 [−23.08, 47.28] |
| Steinmetz 2020 CABG Surgery | 490.2 | 85.2 | 81 | 451.3 | 106.4 | 90 | 12.7% | 38.90 [10.13, 67.67] |
| Ferreira 2020 Lung Resection | −12.1 | 76 | 52 | −16.7 | 56 | 43 | 13.1% | 4.60 [−21.99, 31.19] |
| Liu 2020 Lobectomy Surgery | 21.5 | 43.3 | 37 | −36.1 | 44.4 | 36 | 14.3% | 57.60 [37.48, 77.72] |
| | | | | | | | | |
| Total (95% CI) | | | 409 | | | 389 | 100.0% | 34.41 [12.81, 56.02] |

Heterogeneity: Tau² = 744.05; Chi² = 28.78, df = 8 (P = 0.0003); I² = 72%
Test for overall effect: Z = 3.12 (P = 0.002)

**Fig 4. Mean difference in 6MWT from baseline to approximately 4 weeks post-surgery.**

21.2 to 42.4, $I^2$ 55% $Tau^2$ 221.27 P<0.001) was found in favour of the intervention group. In a sensitivity analysis exploring only Prehabilitation interventions for lung cancer surgery and colorectal surgery there was no change in the overall results. The duration of the interventions ranged from one week to 16 weeks, with three trials not reporting duration and some reporting a range, thus limiting our ability to examine the duration of intervention on outcomes.

A meta-analysis of nine trials [37,47,49,56–58,60,102,105] (n = 798) assessing post-surgery 6MWT at four-week follow-up found a mean difference of 34.4m (95%CI 12.8 to 56.0, $I^2$ 72% Tau2 744.05, P = 0.002) (**Fig 4**). In a sensitivity analysis, we removed the trial by Minnella et al. [57] with a follow-up ranging from 4–8 weeks and the results did not change. There were four trials (n = 305) in the meta-analysis at eight weeks follow-up and there was no longer a significant difference (15.8 m, 95% CI -8.8 to 40.3, $I^2$ 62%, $Tau^2$ 354.89) (S3 Fig in S1 File). The one trial [67] with follow-up up to 26-weeks after surgery found no between-group difference.

**Quality of life.** QoL was assessed in 34 trials [36,39,46,48–50,58,60,62,65,67,69–73,76–78,85–87,90,91,94–96,98,100–102,105,110,112], most commonly through the 36-Item Short Form Health Survey (SF-36). Of 21 trials comparing between-group QoL before surgery, two trials (9.5%) [69,105] found that QoL was significantly greater in the Prehabilitation group across all questionnaire subscales, while seven trials (33.3%) trials found mixed evidence varying by subscale and/or questionnaires [36,39,46,65,67,86]. Most trials that used SF-36 to measure QoL reported the outcomes by the physical component summary score (PCS) and mental component summary score (MCS). There were 11 trials [46,48,50,58,65,67,73,85,91,98,101] (n = 1167) included in the PCS meta-analysis and there were no significant differences between intervention and comparator groups before surgery (mean difference 1.1, 95% CI -0.07 to 2.4, $I^2$ 30%, Tau 1.12, p = 0.07) (S4 Fig in S1 File). Twelve trials [46,48–50,58,65,67,73,85,91,98,101] (n = 1185) were included in the MCS and there were no significant differences between groups (mean difference 0.08, 95% CI –1.3 to 1.4, $I^2$ 14%, $Tau^2$ 0.74, p = 0.91) (S5 Fig in S1 File).

There were eight trials that measured quality of life using the SF-36 post surgery and time points varied from discharge of surgery to one year, thus, it was not possible to conduct a meta-analysis. Descriptively, within one month after surgery, one of seven trials [49] (14.3%) found a favourable effect of Prehabilitation among some questionnaire subscales and one trial (14.3%) found varying results by questionnaire scale [90]. Up to 12-weeks after surgery, one of eight trials [102] found a significant difference favouring Prehabilitation, while one of seven [49] found varying results. Up to 26 weeks after surgery, three of four trials found some evidence supporting Prehabilitation improving quality of life [39,46,90]. No trials with one- [87,98] or two-year [76] follow-up after surgery found between-group differences in QoL.

**Anthropometry.** Of the six trials assessing anthropometry [67,69,76,78,81,85], one of these [76] had only data at two years post-surgery. BMI was measured at pre-surgery and the mean difference was –0.9 kg/m² (95% CI –1.82 to 0.01, p = 0.05, $I^2$ 73%, $Tau^2$ 0.58) in favour of the intervention group (S6 Fig in S1 File). Post-surgery, no between-group differences in anthropometry were observed at 2-year follow-up [76].

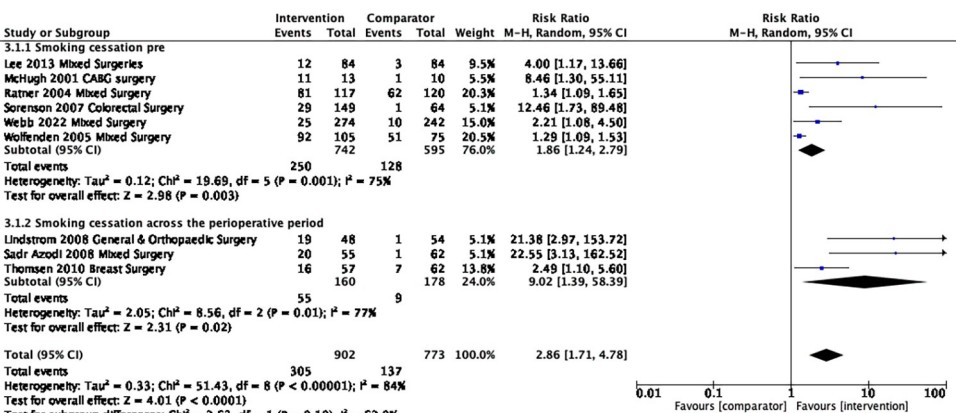

**Fig 5. Risk Ratio of smoking cessation at pre-surgery and across the peri-operative period.**

**Smoking behaviour.** Smoking behaviour was assessed in 17 trials [42–44,54,55,59,63,67–69,79,89,99,103,104,106,107,111] through various self-report methods (e.g., questionnaires, interviews), with ten studies [42–44,54,59,63,103,104,106,107] using biochemical validation (e.g., exhaled CO, urinary cotinine) at least at one-time point in at least a proportion of the participant sample. There were nine trials (n = 1675; no feasibility/pilot studies) [42–44,54,59,69,89,103,107] included in the meta-analysis of smoking abstinence at pre-surgery and the risk ratio was 2.9 (95% CI 1.7 to 4.8, $I^2$ 84%, Tau2 0.33, P <0.001) significantly in favour of the intervention group (**Fig 5**). Three trials [42–44] measured smoking abstinence across the perioperative period i.e., from pre-surgery up to three-weeks post-surgery; in sub-group analysis there were no significant differences.

There were six trials [43,44,55,59,63,106] (n = 991) included in the meta-analysis at 12 months for the abstinence outcome and the risk ratio was 1.74 (95% CI 1.20 to 2.55, $I^2$ 43%, Tau$^2$ 0.09, p = 0.004) (Fig 6).

**Physical activity.** Physical activity was assessed in 19 trials [47,48,50,52,58,60–62,69,70,74,78,82,86,87,90,91,94,98,108,110,111] through various objective (e.g. accelerometry), and subjective methods (e.g. self-report questionnaires and diaries). Of the eight trials assessing between-group differences pre-surgery, five (62.5%) trials [52,69,82,90,98] observed greater physical activity in the Prehabilitation group, while one trial [47] found that Prehabilitation benefited some components of physical activity (i.e., moderate to vigorous physical activity (MVPA)) but not others. Within one-month of surgery, one of six trials [16.7%] found significantly greater physical activity levels in the Prehabilitation group. Up to 12 weeks after

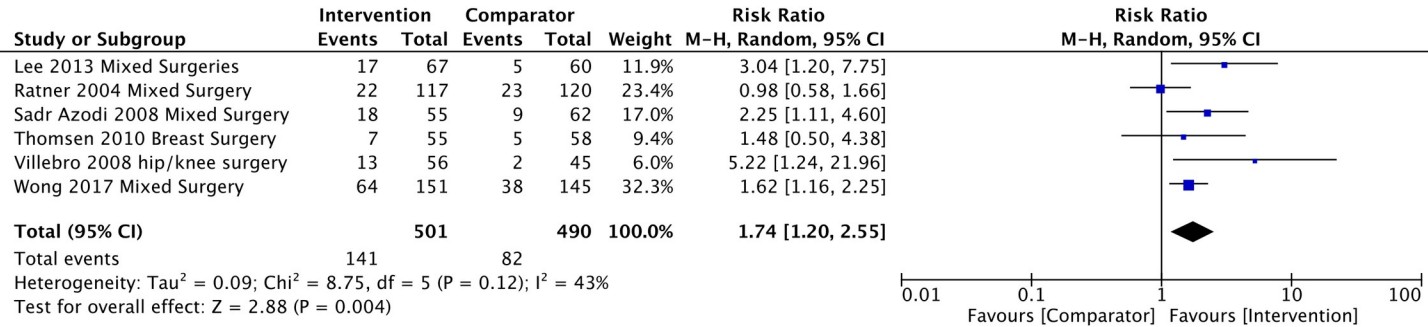

**Fig 6. Risk ratio of smoking cessation at 12-months post-surgery.**

surgery, one trial of three [47] found a significant group difference that favoured Prehabilitation, although this was only for MVPA [47]. One of two trials [90] assessing physical activity at 6-months post-surgery observed a between-group difference that favoured the Prehabilitation group. One [98] of two trials with 12-month follow-up after surgery found a significant effect favouring Prehabilitation

**Dietary intake.**   Two trials assessed dietary intake [52,78]; one through a 2-day dietary recall [78], and the other through a 3-day recall [52]. The one trial that assessed between-group difference in intake before surgery [78] found that daily energy intake was significantly lower in the Prehabilitation group. Between-group difference in dietary intake post-surgery was not assessed in either study.

**Alcohol use.**   Alcohol use was assessed in two studies, one via the AUDIT-C [72] and one via a bespoke questionnaire [111]. Both were pilot and/or feasibility trials and, therefore, between-group comparisons are not summarised here.

**Health inequalities.**   None of the trials included in this review examined differential effects by socioeconomic characteristics, nor did any specifically target lower sociodemographic groups (e.g., lower income patients). Therefore, we were unable to examine the impact of behavioural Prehabilitation interventions on health inequalities.

**Publication bias.**   There was no evidence of publication bias in the studies included in the meta-analyses (S7-S15 Figs in S1 File).

**Risk of bias.**   There were eight trials that were considered high risk of bias, 48 as unclear risk of bias and 11 as low risk of bias (Fig 7).

## Discussion

Prehabilitation interventions were effective at optimizing functional capacity, and smoking cessation prior to surgery. These improvements may have contributed to an average shorter length of stay (-1.5 days) which was observed in the Prehabilitation group, but this is likely to be specific to Prehabilitation for lung cancer surgery as found in sensitivity analysis. There was no evidence that Prehabilitation interventions improved QoL, or reduced BMI just before surgery, although the BMI outcomes were only reported by a small number of trials (n = 4). Regarding post-surgery outcomes, improvements in physical function were sustained for up to four weeks post-surgery. Only smoking cessation data at 12 months could be quantitatively synthesised; greater rates of smoking cessation in the prehabilitation group were observed. That improvements in smoking outcomes were sustained at 12-months post-surgery suggests that the surgical encounter holds promise as a teachable moment for longer-term behavioural change. No studies reported outcomes by health inequality measures, therefore, we could not determine whether Prehabilitation interventions are equitable.

Prehabilitation reduced mean LOS by 1.5 days, similar to findings of another systematic review [113] which found a reduction of 1–2 days compared in patients undergoing joint surgery. However, another systematic review only found a reduction of -0.27 days in colon and rectal cancer patients [114]. In our sensitivity analysis we found that for patients receiving lung surgery LOS was reduced by 3.5 days. There were not more than two surgical specialties and therefore other specialties could not be compared. This suggests that the effect of Prehabilitation differs across surgical procedures and should be investigated in future research. Additionally, we found that Prehabilitation improved 6MWT both before surgery (32m) and after (38m) both exceeding the upper range of the minimal clinically important difference of 30.5 m [115]. Thus, Prehabilitation may improve fitness to undertake the surgery.

Prehabiliation improved rates of smoking cessation before surgery. Research suggests that at least four weeks of successful smoking cessation can reduce respiratory complications, while

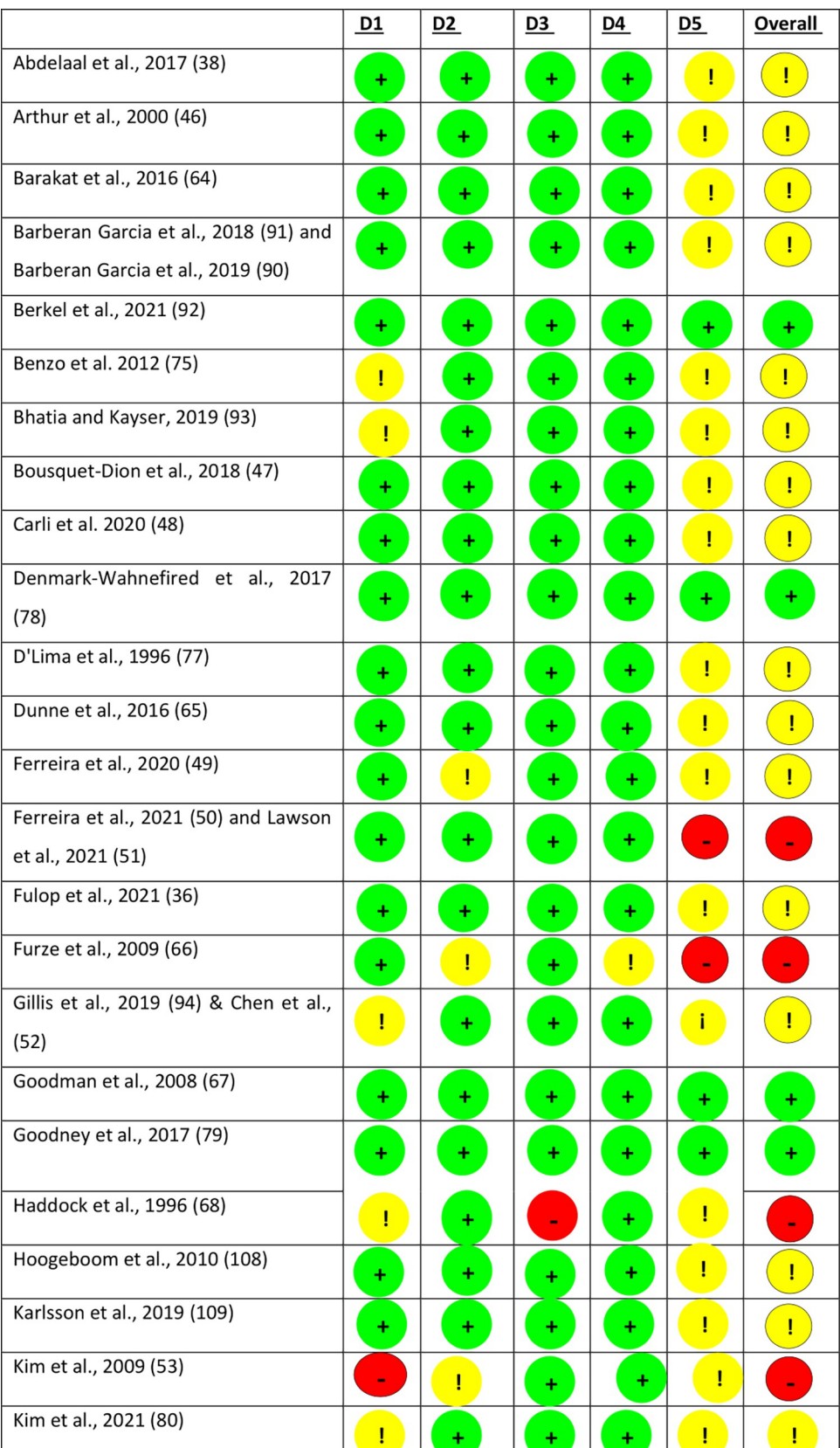

| | D1 | D2 | D3 | D4 | D5 | Overall |
|---|---|---|---|---|---|---|
| Abdelaal et al., 2017 (38) | + | + | + | + | ! | ! |
| Arthur et al., 2000 (46) | + | + | + | + | ! | ! |
| Barakat et al., 2016 (64) | + | + | + | + | ! | ! |
| Barberan Garcia et al., 2018 (91) and Barberan Garcia et al., 2019 (90) | + | + | + | + | ! | ! |
| Berkel et al., 2021 (92) | + | + | + | + | + | + |
| Benzo et al. 2012 (75) | ! | + | + | + | ! | ! |
| Bhatia and Kayser, 2019 (93) | ! | + | + | + | ! | ! |
| Bousquet-Dion et al., 2018 (47) | + | + | + | + | ! | ! |
| Carli et al. 2020 (48) | + | + | + | + | ! | ! |
| Denmark-Wahnefired et al., 2017 (78) | + | + | + | + | + | + |
| D'Lima et al., 1996 (77) | + | + | + | + | ! | ! |
| Dunne et al., 2016 (65) | + | + | + | + | ! | ! |
| Ferreira et al., 2020 (49) | + | ! | + | + | ! | ! |
| Ferreira et al., 2021 (50) and Lawson et al., 2021 (51) | + | + | + | + | - | - |
| Fulop et al., 2021 (36) | + | + | + | + | ! | ! |
| Furze et al., 2009 (66) | + | ! | + | ! | - | - |
| Gillis et al., 2019 (94) & Chen et al., (52) | ! | + | + | + | i | ! |
| Goodman et al., 2008 (67) | + | + | + | + | + | + |
| Goodney et al., 2017 (79) | + | + | + | + | + | + |
| Haddock et al., 1996 (68) | ! | + | - | + | ! | - |
| Hoogeboom et al., 2010 (108) | + | + | + | + | ! | ! |
| Karlsson et al., 2019 (109) | + | + | + | + | ! | ! |
| Kim et al., 2009 (53) | - | ! | + | + | ! | - |
| Kim et al., 2021 (80) | ! | + | + | + | ! | ! |

**Fig 7. Risk of bias assessment for included studies.**

at least three to four weeks can reduce wound healing complications [116]. The duration of several smoking interventions included in this review was less than three weeks, and, therefore, their impact on intra- and peri-operative health outcomes may be limited. We found no difference in rates of smoking cessation at 12-months after surgery. This is similar to findings of a review whereby hospital initiated smoking cessation programmes (in admitted patients) increased smoking cessation for six to 12 months after discharge (risk ratio (RR) 1.37, 95% confidence interval (CI) 1.27 to 1.48; 25 trials) [117]. These effects were produced by high-intensity behavioural interventions that included at least one month of supportive contact after discharge. In times of integrated care, it assumed that community provision has a role to support successful longer-term cessation beyond hospital discharge. Indeed, referral to community smoking cessation services after discharge is a critical component of the evidence-based hospital-initiated tobacco dependency treatment services currently being implemented in the UK as part of the NHS Long Term Plan [118]. However, some studies included in the current review would not be considered high-intensity, and indeed, some consisted of 'one-off' counselling sessions e.g., one 15-minute session, suggesting that lower intensity interventions may still be effective at producing longer-term benefits.

We did not find an improvement in mental or physical QoL in the pre-surgery period. This may be because, while interventions produce significant improvements in some outcomes (functional capacity and smoking), they may not be substantial enough to produce a noticeable effect. For example, an improvement of 32 m in the 6MWT may be clinically significant but may not have a tangible impact on patients' QoL. It may also be those with serious conditions who are experiencing pain and/or discomfort may only achieve improved QoL through surgery. Other reviews that have specifically investigated psychological prehabilitation interventions have found improvements in QoL [119]. Therefore, behavioural interventions alone may not be sufficient to improve QoL in the period leading up to surgery, and psychological support may be necessary.

A systematic review of lifestyle weight loss interventions found that weight loss of 7.2% before surgery in people with obesity can reduce hospital length of stay by 27% [120]. However, most studies included in this review were patients receiving bariatric surgery and these studies were excluded in our review. Intentional weight loss before surgery remains contentious. There are observations of a weight-outcome paradox whereby people with a BMI > 30 $kg/m^2$ appear to experience better outcomes than those with lower BMIs specifically for cardiovascular surgery [121]. One proposed explanation is that plentiful reserves of fat provide energy during periods of accelerated catabolism after major surgery [122], reducing mortality risk. Another is that chronic inflammation, characteristic of obesity, pre-conditions the body against acute excessive inflammation [123], reducing mortality risk. More research is needed to understand under what circumstances (e.g., co-morbidities, baseline BMI, surgery type) weight loss is beneficial, what amount of weight loss is appropriate and at what stage of the surgical pathway.

## Strengths and limitations

This is the first systematic review to consolidate the literature on the characteristics and effectiveness of behavioural prehabilitation interventions targeting health risk behaviours of public health importance (i.e., physical activity, diet (and weight loss), alcohol use and smoking) for improving a range of important outcomes across surgical specialties. As we included unimodal and multimodal interventions, we could not identify which specific intervention components were most effective. While this review focused on four risk behaviours, some of the interventions included other components such as psychological support, protein supplementation and

breathing exercises. This may have influenced outcomes, although, these trials were very few. Around a quarter of trials were either pilot/feasibility trials and only eight (11.9%) were judged to have a low risk of bias, thus overall, there is limited high quality evidence. The risk of bias may have been unclear or high because reporting of intervention and assessment timepoints were not clear; we suspect this may reflect uncertainty around surgery schedules and/or changing health circumstances. Lastly, due to heterogeneity in assessment reporting, some studies were not included in the meta-analyses.

### Unanswered questions and future research

There needs to be better agreement of outcome measures (to synthesise findings) and better reporting including descriptions of the interventions (e.g., use of the TIDiER checklist [124]) to enable service providers to identify the most effective service for their population. Also, the paucity of longer-term post-surgery data means it remains unknown whether Prehabilitation interventions can promote longer-term behavioural change and health improvements. The Prehabilitation literature tends to focus on physiological and mechanistic outcomes with little consideration for the role of behavioral science [125]. To truly leverage surgery as a teachable moment for longer-term outcomes, future interventions must draw on behavioral science [126]. As socioeconomic position is an independent predictor of surgical complications [127,128] examining the effect of Prehabilitation interventions in different socioeconomic groups is important for future research; routine service data may help answer this question rather than trials. Intervention effectiveness across socioeconomic groups is critical in future research investigating digital prehabilitation interventions given poorer digital access and literacy among disadvantaged persons and those with complex needs [129]. Most Prehabilitation interventions included exercise, hence why we found good evidence for improved physical function. However, there were very few studies that focused on alcohol use, dietary intake and weight loss and we did not explore adverse effects of prehabilitation interventions. The cost-efficiency and resource implications of Prehabilitation interventions are an important consideration for hospital systems and should be a priority for future research. If costs associated with delivery of the Prehabilitation service are less than costs saved through bed days released (approximately £342/bed/day in the UK [130]), the service would be 'cost-efficient' [131], and savings could be used to expand provision of Prehabilitation services.

### Conclusions

Behavioural Prehabilitation interventions could be offered to patients across different surgery specialties prior to surgery to help improve functional capacity and smoking cessation outcomes which may enable them to be discharged sooner; however, evidence for shorter length of stay was only observed for patients undergoing lung cancer surgery. That improvements in smoking outcomes were sustained at 12 months post-surgery suggests that the surgical encounter holds promise as a teachable moment for longer-term behavioural change. Given the paucity of data on the effects on other behavioural risk factors, more research grounded in behavioural science and with longer-term follow-up is needed to further investigate this potential.

### Supporting information

**S1 File. Contains all the supporting files, tables and figures.**
(DOCX)

## Author Contributions

**Conceptualization:** Mackenzie Fong, Eileen Kaner, James Prentis, Claire D. Madigan.

**Data curation:** Mackenzie Fong, Maisie Rowland, Henrietta E. Graham, Louise McEvoy, Kate Hallsworth, Gabriel Cucato, Carla Gibney, Martina Nedkova, Claire D. Madigan.

**Formal analysis:** Mackenzie Fong, Henrietta E. Graham, Claire D. Madigan.

**Methodology:** Mackenzie Fong, Eileen Kaner, Maisie Rowland.

**Writing – original draft:** Mackenzie Fong, Claire D. Madigan.

**Writing – review & editing:** Eileen Kaner, Maisie Rowland, Henrietta E. Graham, Louise McEvoy, Kate Hallsworth, Gabriel Cucato, Carla Gibney, Martina Nedkova, James Prentis.

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
