## [Decision Letter · Decision Letter 0]

27 Mar 2023

PONE-D-22-34287Full title The effect of preoperative behaviour change interventions on pre- and post-surgery health behaviours, health outcomes, and health inequalities in adults: a systematic review and metaanalysesPLOS ONE

Dear Dr. Fong,

Thank you for submitting your manuscript to PLOS ONE. After careful consideration, we feel that it has merit but does not fully meet PLOS ONE’s publication criteria as it currently stands. Therefore, we invite you to submit a revised version of the manuscript that addresses the points raised during the review process.

The manuscript has been evaluated by two reviewers, and their comments are available below. They found your study interesting and well conducted overall, but they also raised several concerns that need attention. For instance, they requested clarifications on the methodological reporting of your systematic review and the rationale of the study. Please carefully address all concerns raised.  

We look forward to receiving your revised manuscript.

Kind regards,

Dario Ummarino, PhD

Senior Editor

PLOS ONE

Journal Requirements:

“This study was funded by the NIHR Applied Research Collaboration North East and North Cumbria. This research was supported by the NIHR Leicester Biomedical Research Centre. The views expressed are those of the author(s) and not necessarily those of the NHS, the NIHR, or the Department of Health and Social Care.”

6. Please note that supplementary tables should be uploaded as separate "supporting information" files.

7. Please remove your figures from within your manuscript file, leaving only the individual TIFF/EPS image files, uploaded separately. These will be automatically included in the reviewers’ PDF.

Reviewers' comments:

Reviewer's Responses to Questions

**Comments to the Author**

1. Is the manuscript technically sound, and do the data support the conclusions?

Reviewer #1: Partly

Reviewer #2: Yes

2. Has the statistical analysis been performed appropriately and rigorously? 

Reviewer #1: Yes

Reviewer #2: Yes

3. Have the authors made all data underlying the findings in their manuscript fully available?

Reviewer #1: Yes

Reviewer #2: Yes

4. Is the manuscript presented in an intelligible fashion and written in standard English?

Reviewer #1: Yes

Reviewer #2: Yes

5. Review Comments to the Author

Reviewer #1: Line 73: page 2 - four behavioural health risks - please describe these.

Line 78-79: Please explain the sentence "Behavioural modification.... with health inequalities - is there a ref for this?

Line 89 - 92 - is a very general statement - this review would not be able to succinctly provide this information. My suggestion is to tone the message down.

The current aim of the paper is not clear - is it for health inequalities or health risk behaviours? Please clarify and discuss this appropriately.

Explain why the updated search was only done on Medline and not the other databases too.

In the methods section - please indicate all the reviewers’ names for all searches, screens and conflict resolution.

Line 150; Explain LOS, 6 MWT, QOL, and BMI - these are abbreviations.

Under study characteristic – list refs. (N=19), N=28

Same for the intervention characteristics – list the ref of the articles discussed

Line 402 – explain why Weight loss was discussed when it was not one of the outcomes defined in the health behaviour list.

Ref 32 – 2 full stops in the ref format.

Supplementary table 1 – for the weight loss portion the Amendment and the rationale do not match – the amendment suggested looking at intervention promoting weight gain but the rationale suggests it was done otherwise. Please clarify.

Sup table 2 – NACRT not defined in legends

Reviewer #2: This is a very interesting systematic review (SR) and metaanalysis (MA) about health behavioural interventions delivered during prehabilitation across different surgical specialties. The SR is well conducted and report some interesting findings and the manuscript is very well written and easy to read. I do have, however, some comments that I would like the authors to clarify:

-the search was conducted more than 6 months ago. With more than 20 papers being published about prehabilitation each month, I believe the search should have been updated. At least, monthly alerts should have been set to be up to date in the latest publications. As this is not reported in the methods, I wonder whether this was the case for this review.

-the range for duration of intervention was from one week to 9 months. Was this taken into consideration when performing the meta-analyses? Duration of intervention is important in behavioural interventions such as dietary intake and physical activity, thus this should be taken into account when performing the MA beyond the random model. Have the authors conducted any subanalysis by duration of prehab? This might be interest in some outcomes like LOS, 6MWT or BMI and may reduce heterogeneity when present.

-Although the type of surgery is indicated in the description of the trials, I also think this information should be included somehow in the metaanalysis. I've noticed that most studies included in the MA for LOS were performed in patients undergoing lung cancer surgery. As for duration of intervention, type of surgery night influence the results of prehabilitation on LOS, as some in some surgeries the impact of nutrition and functional capacity might be greater than in others to reduce LOS.

-I would like the authors to clarify what they consider Physical Activity in this paper, because I feel the SR includes a mix of exercise-based interventions (some supervise) and physical activity promotion which are two different things that lead to different results. Could the authors please elaborate in the methods section what do you consider by a physical activity intervention during prehabilitation?

-There is one subanalysis from the main study by Barberan-Garcia et al. that was not included in the review and should have been, as it includes information regarding post-surgery functional capacity and physical activity levels. The reference for the study is Barberan-Garcia et al. BJA 2019, 123(4):450-456.

-Why only studies that reported LOS in mean and not median were included in the MA? As you know, mean and SD can be calculated from median and 25-75 quartiles.

6. PLOS authors have the option to publish the peer review history of their article (what does this mean?). If published, this will include your full peer review and any attached files.

Reviewer #1: No

Reviewer #2: No

---

## [Author Response · Author response to Decision Letter 0]

15 May 2023

Response to reviewers has been uploaded in the 'cover letter' file.

---

## [Editor Report · Decision Letter 1]

23 May 2023

Full title The effect of preoperative behaviour change interventions on pre- and post-surgery health behaviours, health outcomes, and health inequalities in adults: a systematic review and metaanalyses

PONE-D-22-34287R1

Dear Dr. Fong,

We’re pleased to inform you that your manuscript has been judged scientifically suitable for publication and will be formally accepted for publication once it meets all outstanding technical requirements.

Kind regards,

Favil Singh

Guest Editor

PLOS ONE

Additional Editor Comments (optional):

The final files that you submitted for your manuscript have been checked and have been found to be suitable.
---

## [Editor Report · Acceptance letter]

26 Jun 2023

PONE-D-22-34287R1 

The effect of preoperative behaviour change interventions on pre- and post-surgery health behaviours, health outcomes, and health inequalities in adults: a systematic review and meta-analyses 

Dear Dr. Fong:

I'm pleased to inform you that your manuscript has been deemed suitable for publication in PLOS ONE. Congratulations! Your manuscript is now with our production department. 

Kind regards, 

on behalf of

Dr. Favil Singh 

Guest Editor

PLOS ONE